# Viral Shedding in Mice following Intravenous Adenovirus Injection: Impact on Biosafety Classification

**DOI:** 10.3390/v15071495

**Published:** 2023-07-01

**Authors:** Christopher J. LaRocca, Kari L. Jacobsen, Kazuho Inoko, Stanislav O. Zakharkin, Masato Yamamoto, Julia Davydova

**Affiliations:** 1Department of Surgery, University of Minnesota, Minneapolis, MN 55455, USA; 2Masonic Cancer Center, University of Minnesota, Minneapolis, MN 55455, USA; 3WebMD, New York, NY 10014, USA; 4Institute of Molecular Virology, University of Minnesota, Minneapolis, MN 55455, USA

**Keywords:** adenovirus, biosafety, viral shedding, biosafety level (BSL), animal models

## Abstract

There have been numerous advances in gene therapy and oncolytic virotherapy in recent years, especially with respect to cutting-edge animal models to test these novel therapeutics. With all of these advances, it is important to understand the biosafety risks of testing these vectors in animals. We performed adenovirus-based viral shedding studies in murine models to ascertain when it is appropriate to downgrade the animals from Biosafety Level (BSL) 2 to BSL 1 for experimental handling and transport. We utilized intravenous injections of a replication-competent adenovirus and analyzed viral shedding via the collection of buccal and dermal swabs from each animal, in addition to obtaining urine and stool samples. The adenovirus hexon copy number was determined by qPCR, and plaque formation was analyzed to assess the biologic activity of viral particles. Our results demonstrate that after 72 h following viral inoculation, there is no significant quantity of biologically active virus shedding from the animals. This observation suggests that on day 4 following adenovirus injection, mice can be safely downgraded to BSL 1 for the remainder of the experiment with no concern for hazardous exposure to laboratory personnel.

## 1. Introduction

The fields of gene therapy and oncolytic virotherapy have made remarkable progress in recent years with respect to vector design, transgene delivery, and the development of novel combination regimens, which have resulted in marked improvements in the therapeutic potential of these platforms [1]. Oncolytic viruses can be optimized to confer cancer-specific replication while minimizing off-target effects on normal tissues, and they can be attenuated to minimize pathogenicity, all towards the goal of improving safety and tolerability in clinical settings [2].

For oncolytic virotherapy researchers, relevant and cutting-edge animal models are essential for the generation of meaningful preclinical data to assess the potential for clinical translation [3]. Understanding the biosafety risks of testing these vectors in animals is critically important for both institutions and laboratory personnel, especially when it comes to viral shedding following initial inoculation [4].

Oncolytic adenoviruses are one of the most common and well-studied viral vector platforms in gene therapy and virotherapy [5]. There are multiple characteristics of adenoviruses that make them particularly applicable to gene therapy approaches: they have an excellent safety profile (infection typically results in “common cold” symptoms), they can infect both dividing and non-dividing cells, they do not incorporate into the host genome, and they do not result in mutagenesis/oncogenesis. Furthermore, the 35 kb double-stranded DNA genome is easily modified and has a large transgene capacity, which can be utilized to include cytokines, reporter genes, or other therapeutic transgenes in the viral genome [6,7,8,9,10].

The importance of animal model selection is paramount to the success of pre-clinical research as investigators are challenged to find the best model that recapitulates the tumor microenvironment of a specific malignancy, and it facilitates the testing of novel therapeutics. These points are especially relevant for those working with replication-competent adenoviral vectors as a well-known challenge to animal models for testing adenoviral therapeutics is that there is limited adenoviral replication in murine tissues [11,12,13]. Another model system that is often utilized is the Syrian Golden Hamster, which has the advantages of a competent immune system and is semi-permissive to adenovirus replication [14]. Furthermore, another area of investigation is the development of a swine model, which has thus far been shown to support adenovirus replication [15]. However, to date, immunodeficient mice bearing human tumor xenografts are the most used models as they allow to test the efficacy of adenovirus-based vectors in human tumor tissues. 

At all research centers, Institutional Biosafety Committee (IBC) practices are key for ensuring the safety of laboratory personnel who may be working with viral vectors. The National Institutes of Health (NIH) Biosafety Guidelines designates biosafety levels (BSL) for viral vectors depending on their unique properties and potential for side effects if there is an unplanned exposure. Some of the key questions/concerns that are often raised are the degree of viral shedding in animal models and when it is safe to handle/transport these research animals following viral infection [16].

In this manuscript, we report our findings from adenovirus-based viral shedding studies in murine models after infection with human adenovirus serotype 5 to ascertain when it is appropriate to downgrade the animals from BSL 2 to BSL 1 for experimental handling and transport.

## 2. Materials and Methods

### 2.1. Cell Lines and Culture Conditions

The human lung adenocarcinoma cell line (A549) was obtained from the American Type Culture Collection (ATCC). The transformed human embryonic retina cell line 911 was a gift from Dr. Alex J. van der Eb (Leiden University, Leiden, The Netherlands). Cells were cultured in Dulbecco’s Modified Eagle Medium (DMEM) and supplemented with 5% fetal bovine serum and a 1% penicillin–streptomycin mixture. The cells were maintained as monolayers in a humidified incubator with 5% CO_2_ at 37 °C.

### 2.2. Viral Vectors

The adenovirus used in the experiment is a human wild type adenovirus serotype 5 (hAd 5) with intact E1, E2, and E3 regions and capsid. It did not have any genomic deletions or insertions. This vector has been previously used as wild-type replication competent control vector [9,17,18].

### 2.3. In Vivo Murine Model

Female athymic nude mice (Crl:NU(NCr)-Foxn1nu, Charles River Laboratories, 8 weeks old) were divided into two cohorts of five animals for tumor- and non-tumor-bearing groups. Prior to initiation of this protocol, athymic nude mice were housed in microisolator caging and maintained under specific pathogen-free (SPF) conditions. 

A549 cells, commonly used as an hAd 5 replication permissive and coxsackie-adenovirus receptor (CAR) positive control, were used to establish bilateral subcutaneous tumors in the flanks (2 × 10^6^ cells in 100 µL of PBS) in half of the mice (tumor-bearing group) [19]. The other half of mice was tumor-free as our goal was to evaluate virus shedding in both model systems.

When the tumors in the tumor-bearing group reached approximately 6–8 mm in diameter, all mice were moved to BSL 2 housing. The mice were injected intravenously into the tail vein with 5 × 10^10^ viral particles of hAd 5 (diluted in 100 µL of sterile PBS). 

The viral copy numbers in samples collected from both tumor-bearing and tumor-free groups were compared to the site-specific baseline levels obtained prior to viral injection (day 0, please see below “Sample Collection”). 

All procedures were carried out according to protocols approved by the Institutional Animal Care and Use Committee (IACUC) of the University of Minnesota.

### 2.4. Sample Collection in Mice

Buccal, dermal, urine, and fecal samples were collected on day 0 (prior to adenovirus injection) and on days 1, 3, 4, and 7 after viral injection. Buccal swabs were collected by gently brushing the interior buccal surfaces and tongue. Dermal swabs were collected by brushing the dorsal flanks in the area overlying the established tumors. Urine swabs were collected by gently stimulating the urethral opening, thereby causing the animals to urinate. Samples were collected on sterile cotton swabs, placed in sterile microtubes, immediately frozen in a dry ice/ethanol slurry, and then transferred to −80 °C for storage. Fecal pellets were collected directly from animals when produced during handling and similarly frozen in sterile microtubes until further processing.

Note: Our original experimental design included time points at only days 0, 1, 3, and 7. However, when we observed viral copy numbers and plaques at day 3 following viral infection, we added a day 4 time point in a separate experiment. Thus, a total of 15 mice bearing tumors on both flanks were utilized in all experiments. Five mice were available at each time point on days 0, 1, 3, and 7. For day 4, the tumor-free group had three mice, and the tumor-bearing group had two mice. Data have been combined and presented on the same figures.

### 2.5. DNA Extraction from Buccal, Dermal, Urine, and Fecal Samples

Frozen samples were stored at −80 °C until use. Buccal and dermal samples were processed using the Qiagen QIAamp DNA Blood Mini Kit, urine samples were processed using the QIAGEN QIAamp Viral RNA Mini Kit, and fecal samples were processed using the QIAGEN QIAamp DNA Stool Kit, according to manufacture protocol. Buccal and dermal swabs were vortexed while submerged in PBS, per manufacture protocol, to release sample particles from cotton. Urine swabs were submerged and vortexed in 500 µL of PBS and split into one volume of 140 µL for use in qPCR and one volume of 360 µL for use in PFU analysis. Buccal and dermal samples were eluted in final volumes of 100 µL. Fecal samples were eluted in 200 µL, and urine samples were eluted in 60 µL.

### 2.6. qPCR Analysis to Evaluate Viral Copy Numbers

qPCR was performed using SYBR Green PCR Master Mix (Applied Biosystems catalog # 4309155) in 20 µL sample volumes according to the manufacturer’s instructions. Viral hexons were amplified individually with the following primers: hexon–[(s)–GACCGCAGTTGACAGCATTA and (as)–ATAAAGAAGGGTGGGCTCGT] [19,20]. Quantification was performed in triplicate using Roche LightCycler 480 Instrument II 384-well platform. The data shown are the average values for technical replicates. 

### 2.7. Confirmation of Virus Biological Activity by Plaque Formation Assay

Overall, 911 cells were plated at a density of 1 × 10^6^ cells per well in six-well plates and allowed to adhere overnight. An amount of 800 µL of complete media (Corning DMEM + 5% FBS, 1% Pen/Strep/AmphB) was added to tubes containing sample swabs and vortexed thoroughly. Samples were snap frozen in a dry ice and ethanol slurry and then thawed and vortexed; this was repeated three times. Swabs were removed, and samples were centrifuged for 5 min at 5000 rpm in a benchtop centrifuge to pellet debris. The supernatant was filtered through a 0.22 um syringe filter into a fresh, sterile tube. Any samples that had lost volume via absorption by swab or filter membrane were brought up to 650–700 µL to ensure sufficient volume for the assay. Each sample was serially diluted 1:10 five times, and 500 µL of each dilution was used to infect one well of previously prepared 911 cells. After two hours’ incubation, infection media was aspirated and replaced with 0.5% agar in DMEM +15% FBS, 1% pen/strep/AmphB, followed by an overlay of complete media. Plates were incubated at 37 °C with 5% CO_2_ throughout. Plaque formation was monitored for 9–11 days, and the final plaque appearance was counted when plaques were individually identifiable [21]. A Leica DMi1 microscope was used to monitor the plaques, and they were photographed with a Leica MC120 HD camera. Plaques were sterilely collected by punch method, and DNA was extracted using QIAGEN QIAquick Gel Extraction Kit, according to manufacturer protocol. PCR amplification was performed using QIAGEN Fast Cycling PCR Kit according to manufacturer instructions. Viral primer sequences are AB-1 (AB loop of Ad5) [(s)–GACCACACCAGCTCCATCTC, (as)–GGGTCCAGGAAGGAATTGTT]. Electrophoresis was run on 2% TAE gel with 10 µL of PCR product.

### 2.8. Statistical Methods

Differences in viral replication and copy number were analyzed using the non-parametric Wilcoxon signed-rank test, which does not have distribution assumptions. Statistical comparisons were performed in R 4.2.2. 

Comparisons with the baseline (mice prior to adenovirus inoculation, day 0) were conducted using organ-specific combined tumor and non-tumor baseline values, which were not significantly different from each other. The results were considered statistically significant when the *p*-value was less than 0.05. Data are expressed as a mean ± standard deviation unless otherwise noted.

## 3. Results

While immunodeficient mice bearing human cancer xenografts are commonly used to evaluate the therapeutic potential of adenovirus-based vectors, it is known that human adenovirus does not replicate in murine tissues [11,12,15]. As part of the experimental design, half of the animals had replication-permissive human A549 subcutaneous xenografts, while the other half of the animals remained tumor-free as we wanted to evaluate virus shedding in both model systems.

To demonstrate the degree of viral shedding following intravenous adenovirus injection, buccal and dermal swabs were obtained from each animal, in addition to the collection of urine and stool samples. The adenovirus hexon copy number was determined by qPCR, and plaque formation was analyzed to assess the biologic activity of viral particles. 

Experimental values for viral copy number were compared to the site-specific baseline levels (obtained on day 0 prior to viral inoculation). 

### 3.1. Evaluation of Viral Copy Number and Biologic Activity in Murine Buccal Samples

For the buccal samples (from tumor-bearing mice), the highest mean hexon copy number value (1.3 × 10^4^) was obtained on day 1 following viral infection. There was a marked drop-off in the hexon copy number on days 3 (7.9 × 10^3^) and 4 (6.6 × 10^2^). While values for days 1 and 3 demonstrated statistically significant differences from the baseline (*p* < 0.05), there were no significant differences from the baseline on days 4 and 7 (Figure 1A).

In the “tumor-free” buccal group, the highest mean hexon copy number value (8.0 × 10^4^) was obtained on day 1 following viral infection. Similarly to the above trend, there was a precipitous drop in hexon copy number values on subsequent days following viral infection. Once again, days 4 and 7 demonstrated hexon copy number values that were not statistically significantly different from the site-specific baseline (Figure 1A).

Positive values in viral copy number produced by PCR analysis do not necessarily represent biologically active virus particles (as fragments of viral particles or defective viral particles that cannot infect cells can be captured by this assay) [21]. Therefore, plaque formation analysis was utilized to assess for a biologically active virus (Figure 2).

In the tumor-bearing mice, the buccal samples resulted in plaque formation only at days 1 and 3 following viral infection (Figure 1C). In each case, there was only one plaque that formed. In the mice without tumors, the buccal samples only resulted in plaque formation on day 1, which was not statistically significant compared to day 0. Importantly, there were no plaques beyond day 3 in any buccal samples, signifying that there was no active virus beyond 72 h following viral inoculation.

### 3.2. Evaluation of Viral Copy Number and Biologic Activity in Murine Dermal Samples

For the dermal samples (from tumor-bearing mice), the highest mean hexon copy number value (1.1 × 10^5^) was obtained on day 1 following viral infection. As seen in the buccal samples, there was a steep drop in hexon copy numbers on days 3, 4, and 7 with all of these values being on the order of 10^3^. Hexon copy number values for day 1 and day 3 were significantly different from the baseline (*p* < 0.05), but this trend did not persist at later time points (Figure 1B).

In the non-tumor-bearing mice, the highest mean hexon copy number value (1.0 × 10^5^) was on the first day after virus administration with gradual declines on day 3 (4.5 × 10^4^), 4 (1.6 × 10^4^), and 7 (2.3 × 10^3^) after inoculation. Days 1, 3, and 4 demonstrated values that were statistically significantly different than the site-specific baseline (*p* < 0.05) (Figure 1B). There was no difference between day 7 and baseline values.

For the dermal plaque formation experiment, there is an important contrast between the groups of mice bearing tumors and those without tumors (Figure 1D). The maximum number of plaques in the dermal tumor group on day one was 200, while the maximum number for the dermal-tumor-free group was 10. In the dermal tumor group, the maximum number of plaques on day 3 was two, which decreased to zero by day 4 following viral infection. In the non-tumor group, there was a maximum of five plaques on day 3. 

Interestingly, there was one plaque in the dermal-tumor-free group on day 4. We do not believe that there was a significant quantity of the biologically active virus at this time point, especially given the low viral titer obtained via hexon copy number analysis and on account of a relative comparison to the tumor group, which had no plaques at day 4 (and started with 200 on day 1). Furthermore, the statistical analysis of the plaque assay demonstrated no significant difference between the day 4 value and the baseline value.

### 3.3. Evaluation of Viral Copy Number and Biologic Activity in Murine Urine and Fecal Samples

For the urine samples, viral hexon copy number values for the majority of samples in both the tumor and non-tumor groups were at or below the level of detection of the PCR (Figure 3A). Given such a low level of virus, no plaque formation analysis was performed for the urine samples. This observation fits with observations from other authors who have noted low levels of adenovirus shedding in urine in both human and animal studies, although it may be somewhat dependent upon the route of administration and the dose of initial inoculum [4,15,22,23].

In the fecal samples, there was no difference between the baseline values and the collected time points (Figure 3). For the day 1, 3, and 7 time points, there were no statistically significant differences across both the tumor-bearing and tumor-free mice (*p* > 0.05). As above, given the low viral titers in the fecal samples, no plaque formation analysis was performed.

No experiments were performed on day 4 for urine or fecal samples given such low viral titers, which were near baseline levels on earlier time points.

## 4. Discussion

Oncolytic viruses are emerging as promising therapeutic options for cancer patients with advanced malignancies that are not responding to traditional treatments such as chemotherapy or surgery [24]. The continued clinical translation of these novel therapeutics is of paramount importance to improve cancer outcomes, but there are numerous barriers including regulatory approvals, financial cost, and patient accrual into trials [25]. 

Preclinical research is pushing forward at a rapid pace given the continued advances in gene editing and oncolytic virotherapy. To that end, biosafety is critically important in both the planning and execution of experiments using viral vectors [21,26]. Based on the current NIH Biosafety Guidelines, both replication-deficient and replication-competent adenoviruses belong to Risk Group 2 (RG2), which mandates Biosafety Level (BSL) 2 in laboratory and animal experiments [21,27]. While the specific details of biosafety requirements may vary slightly between institutions, a general requirement is that all experiments with biohazard materials must receive approval from the Institutional Biosafety Committee (IBC). Before this study, in our institution (University of Minnesota, Minneapolis, MN, USA), all experiments with replication-competent adenoviruses must be performed in a Biosafety Level (BSL) 2 setting for the entire period of time until the virus is inactivated (e.g., by oxidation with bleach or fixation with formaldehyde). However, for animal experiments in particular, there are potential logistical, support staff, lab space, and resource concerns that can make conducting prolonged experiments in BSL 2 settings challenging for investigators [16,28]. Given these issues, we aimed to determine whether it would be safe to downgrade animal housing from BSL 2 to BSL 1 following adenovirus inoculation. Interestingly, despite this being a common concern among research groups, there are surprisingly few studies on this topic [4,16].

To ascertain the degree of viral shedding following intravenous adenovirus administration, we utilized murine models (both with and without adenovirus replication-permissive human cancer xenografts) and collected buccal, dermal, urine, and stool samples. Our rationale was that a comparison between naïve animals and those carrying tumors of human origin would provide more valuable information regarding the possibility of increased infection risk associated with adenovirus replication-permissive cells in a murine model. Thus, our studies have utilized A549 human lung adenocarcinoma cells, a cell line in which human adenovirus is highly replicative. We initially hypothesized that if the A549 cells are producing viral progeny, the tumor-bearing group should display an increase in virus recovery compared to the naïve cohort. Interestingly, our studies have demonstrated that the animals bearing A549 human tumors showed no significant difference in virus shedding from the tumor-free animals after 3 days following viral injection. 

In addition to the biology of our chosen murine and tumor models, we have considered details regarding the infectious profile of adenoviral vectors. In determining the potential risk of exposure, we should assume a worst-case scenario. Therefore, we have utilized replication-competent, wild-type human adenovirus with the intact genome (hAd 5) to ensure the maximum virus progeny recovery and maximize our estimation of exposure risk. Notably, the employment of hAd 5, which has historically been a gold standard control for studying adenovirus replication, makes our study significantly different from a previous report that utilized replication-incompetent vectors [4].

Our experiments demonstrated that, following intravenous injection, adenoviral copy numbers quantitated by qPCR were generally found in most of the samples on day 1 after viral administration but quickly reduced to background levels in buccal, urine, and fecal samples. On the other hand, the qPCR analysis indicated that several dermal samples contained high viral counts on day 4 (dermal-tumor-free group), although no hAd 5 biological activities were presented by the PFU assay corresponding to these highest qPCR results.

One of the important distinctions for this type of experimentation is the presence of inactive viral particles or fragments versus those particles that still retain biologic activity. Herein lies the importance of the plaque formation assay to identify biologically active adenovirus [29,30,31]. In all but one of the buccal and dermal samples tested, there was no evidence of any plaques beyond day 3. For that one mouse in the dermal-tumor-free group, a single plaque at day 4 does not represent any clinically relevant volume of biologically active/infectious adenovirus.

Thus, our results demonstrate that, after 3 days following the intravenous injection of a replication-competent adenovirus into murine tail veins, there is no significant quantity of biologically active virus shedding from the animals. This observation suggests that, on day 4 following adenovirus injection, mice can be safely downgraded to BSL 1 for the remainder of the experiment with no concern for hazardous exposure to laboratory personnel. Based upon this data, the University of Minnesota IBC committee changed its policy to support the downgrading of murine animals to BSL 1 after 72 h following adenovirus inoculation. We believe this finding and protocol to be applicable to other researchers working with oncolytic vectors, especially adenovirus-based therapies.

Oncolytic adenoviruses are promising tools in the cancer treatment armamentarium thanks to their ability to not only lyse tumor cells but also initiate a host immune response against the cancer. Overly stringent regulations (such as prolonged containment at a higher BSL level) can serve as an obstacle to the successful clinical translation of these novel therapeutics and impede progress in the field. While our data are limited to adenovirus serotype 5 vectors and murine models, future studies will explore additional viral serotypes and animal models. We hope that our data reported in this manuscript can contribute to facilitating a more efficient clinical translation process of novel viral-based therapeutics.

## Figures and Tables

**Figure 1 viruses-15-01495-f001:**
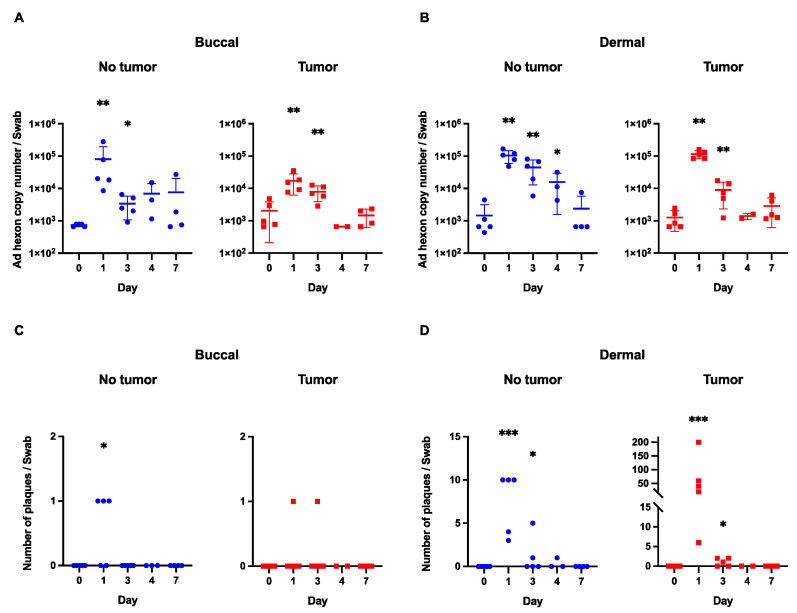
Viral shedding in buccal and dermal samples. (**A**) Viral hexon copy number in buccal samples (both in tumor-bearing and non-tumor-bearing mice) shows a marked decrease with each time point after initial inoculation. (**B**) A similar trend is observed in the dermal samples with a gradual decline in viral copy number following viral injections. (**C**) There were no plaques beyond day 3 in any of the buccal samples. (**D**) With the exception of one plaque from a non-tumor-bearing animal, there were no plaques beyond day 3 after viral infection in the dermal group. *** *p*-value < 0.001; **—*p*-value < 0.01; *—*p*-value < 0.05.

**Figure 2 viruses-15-01495-f002:**
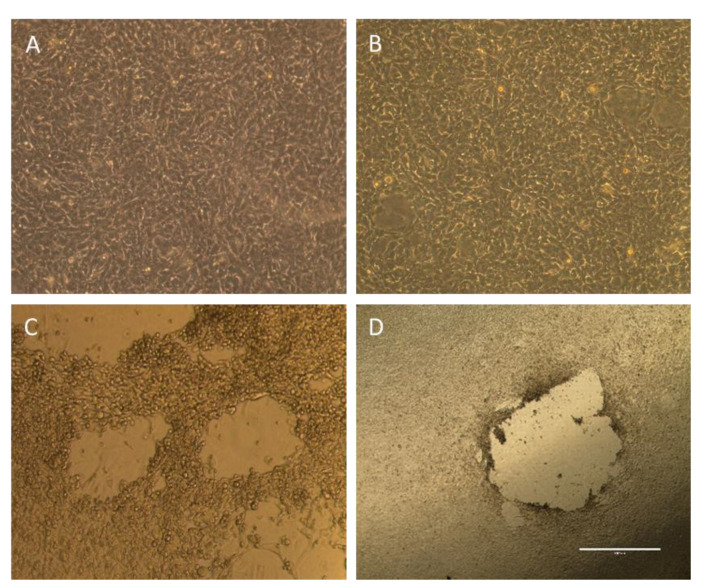
Plaque formation analysis. (**A**,**B**) Plaque formation analysis from buccal and urine swabs demonstrating intact cell monolayer, which signifies that there is no biologically active virus in the samples. (**C**,**D**) Representative images of viral plaques (voids in cell monolayer) signifying a biologically active virus from a positive control infected with hAd 5 (**C**) and a dermal swab (**D**).

**Figure 3 viruses-15-01495-f003:**
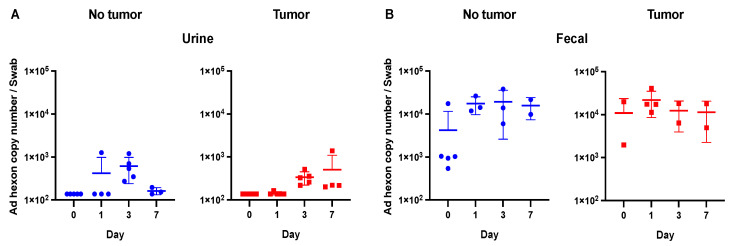
Viral shedding in urine and fecal samples. (**A**) Urine samples were at or below the limit of detection of the qPCR (1 × 10^3^). (**B**) No significant differences were observed in fecal samples between days 1, 3, and 7 and the baseline time point (day 0).

## Data Availability

More details on the data presented in this article are available on request from the corresponding authors.

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
