# Peer review of "Viral Shedding in Mice following Intravenous Adenovirus Injection: Impact on Biosafety Classification"

_viruses, 2023, doi:10.3390/v15071495_

Round 1

Reviewer 1 Report

The manuscript discusses the importance of understanding biosafety risks associated with testing gene therapy and oncolytic virotherapy in animal models. The authors performed Ad5-based viral shedding studies in murine models and found that after 72 hours following viral inoculation, there is no significant quantity of biologically active virus shedding from the animals. This led the authors to conclude that mice can be safely downgraded to BSL1 on day 4 following Ad5 injection.

The manuscript requires minor revisions for publication:

I would edit the first sentence of your intro, as it is identical to the first part of your abstract (“: The fields of gene therapy and oncolytic virotherapy have made remarkable progress in recent years”).

In your introduction, I strongly suggest that you include a few sentences on the murine animal model you use and why you use it – and also provide some general background on animal models used to study (human) adenoviruses.

Line 77f: How many mice did you use in total? Provide information on their age and sex. Did you include control groups (e.g. without tumors, mock infected)?

Line 82: 5x1010 should be 5x1010. Also, see line 118 and elsewhere in your text.

Line 130f: How was the plaque formation monitored? Which scope did you use to make the pictures in Fig. 2? Have these experiments been performed in a blinded manner to avoid any bias? Or did you use a machine to count the plaques for you?

Line 138f: Why do these E4 primer sequences differ from the ones used in the qPCRs in line 108f?

Line 139: What is AB-1?

Line 143f: Stats test should also be performed on the plaque assay data!

Results paragraphs “Buccal”, “Dermal”, “Urine & Fecal”: I suggest you revise these section titles to “Ad5 genome copies and replicating virus in buccal samples” a.s.o… Or similar, more comprehensive paragraph titles.

Fig. 1: Where are the E4 data? You used E4 primers for these experiments, too – you at least stated that in the M&M paragraph. Could you use log scales for the y-axes in panels A and B?

Given the high hexon copy numbers on day 4 in the dermal swabs of the “no-tumor-group” and the plaques, wouldn’t it be safer to say that mice can be safely downgraded to BSL1 on day 7 following Ad5 injection?! 

Fig. 2: How long is the scale bar in D? That information has to be included in the figure caption. To me, it looks like the C and D images were not acquired with the same magnification as A and B... 

Line 231f: It would be good to follow up on the fecal samples, as it seems that you detected continuous shedding. Why not perform plaque assays on these samples to show if there’s replicating virus in there?

Line 236f: Why did you choose to perform day 7 experiments but not on day 4?

Fig. 3: I suggest you show log scales for the y-axes in panels A and B up to 100,000 max. There’s also no need to show y-axes up to 300,000.

Line 280f (and line 191 as well…): “is not felt to represent any clinically relevant volume” is not scientifically accurate. The recommendations should be based on facts, not feelings. You’d be more accurate with your recommendations if you follow up on the fecal samples and state that it’s safe to downgrade to BSL1 at 7 dpi. That should be, at a minimum, discussed!

You should also discuss that these recommendations only apply to mice infected with Ad5 – and that more studies are needed to make this applicable to other replication-competent Ads.

Add a few words on the limitations of your study to the Discussion.

Please also provide an author contribution statement.

Author Response

We thank the reviewers for their time and thoughtful critique of our manuscript entitled Viral Shedding in Mice Following Intravenous Adenovirus Injection:  Impact on Biosafety Classification.  We believe that the critiques have helped us to greatly improve the manuscript.  Below we have outlined individual responses to each of the concerns.

Thank you once again for your time and consideration.  We hope that the editors and reviewers will find this revised manuscript suitable for publication.

Sincerely,

Christopher J. LaRocca, Julia Davydova, and co-authors

Reviewer #1:  The manuscript discusses the importance of understanding biosafety risks associated with testing gene therapy and oncolytic virotherapy in animal models.  The authors performed Ad5-based viral shedding studies in murine models and found that after 72 hours following viral inoculation, there is no significant quantity of biologically active virus shedding from the animals.  This led the authors to conclude that mice can be safely downgraded to BSL1 on day 4 following Ad5 injection.  The manuscript requires minor revisions for publication:

  1. I would edit the first sentence of your intro, as it is identical to the first part of your abstract (“: The fields of gene therapy and oncolytic virotherapy have made remarkable progress in recent years”).

Response:  The abstract has been edited so that the two sentences are no longer identical.

  1. In your introduction, I strongly suggest that you include a few sentences on the murine animal model you use and why you use it – and also provide some general background on animal models used to study (human) adenoviruses.

Response:  The introduction has been augmented with a paragraph that reflects the current models utilized in oncolytic adenovirus-based research.

  1. Line 77f: How many mice did you use in total? Provide information on their age and sex. Did you include control groups (e.g. without tumors, mock infected)?

Response:  A total of 15 female anythmic nude mice (8 weeks old) were used in the experiment.  The mice were divided in two gtoups: As a control, we used the site-specific baseline parameters  obtained prior to viral injections in the same 15 mice.  The methods/results sections were clarified to reflect these points.

  1. Line 82: 5x1010 should be 5x1010. Also, see line 118 and elsewhere in your text.

Response:  The text was reformatted to properly represent the superscripts in their appropriate positions.

  1. Line 130f: How was the plaque formation monitored? Which scope did you use to make the pictures in Fig. 2? Have these experiments been performed in a blinded manner to avoid any bias? Or did you use a machine to count the plaques for you?

Response:  Plaque formation assay is a standard technique for assessing virus biologic activity, and so it is not necessary or appropriate to have the researchers be blinded.  It is routine to count these plaques manually.  The scope was a Leica DMi1 with camera MC120 HD.

  1. Line 138f: Why do these E4 primer sequences differ from the ones used in the qPCRs in line 108f?

Response:  E4 primer description was removed from the manuscript text, as this data was not included in the manuscript.

  1. Line 139: What is AB-1?

Response:  This was clarified in the text.  It is the name of the primer for the AB loop region of Ad5.

  1. Line 143f: Stats test should also be performed on the plaque assay data!

Response:  Statical analysis was performed on the plaque assay data as requested.  Results reflected in the results section.  In short, there were significant differences between baseline values and days 1 and 2, but not at any later time points in the experiment.

  1. Results paragraphs “Buccal”, “Dermal”, “Urine & Fecal”: I suggest you revise these section titles to “Ad5 genome copies and replicating virus in buccal samples” a.s.o… Or similar, more comprehensive paragraph titles

Response:  Paragraph titles were made to be more descriptive per reviewer suggestion

  1. Fig. 1: Where are the E4 data? You used E4 primers for these experiments, too – you at least stated that in the M&M paragraph.

Response:  E4 data is not presented in this manuscript.  To reduce any confusion, the mention of E4 primers in the methods section was removed.

  1. Given the high hexon copy numbers on day 4 in the dermal swabs of the “no-tumor-group” and the plaques, wouldn’t it be safer to say that mice can be safely downgraded to BSL1 on day 7 following Ad5 injection?!

Response:  There are no plaques on day 4 in the dermal “no tumor” group, despite having had numerous plaques at earlier time points.  This observation supports the quick rate at which biologically active virus decreases following injection.  We believe that our results support downgrade to BSL 1 after 72 hours.

  1. Fig. 2: How long is the scale bar in D? That information has to be included in the figure caption. To me, it looks like the C and D images were not acquired with the same magnification as A and B.

Response:  These are representative images, not meant to be directly compared.  Legend modified to clarify.

  1. Line 231f: It would be good to follow up on the fecal samples, as it seems that you detected continuous shedding. Why not perform plaque assays on these samples to show if there’s replicating virus in there?

Response:  There was no significant difference between baseline values and the time points.  Plaque assay were not performed given the absence of any difference.

  1. Line 236f: Why did you choose to perform day 7 experiments but not on day 4? 

Response:  Day 4 experiments were performed on a subset of mice as a separate experiment after identifying plaques at day 3 but not day 7 in the buccal and dermal mice groups.  For urine/fecal, no day 4 was necessary as there was no significant differences in viral copy number between baseline and any of the experimental time points.

  1. Fig. 3: I suggest you show log scales for the y-axes in panels A and B up to 100,000 max. There’s also no need to show y-axes up to 300,000.

Response:  Log scales were included for the figures per the reviewer suggestion.

  1. Line 280f (and line 191 as well…): “is not felt to represent any clinically relevant volume” is not scientifically accurate. The recommendations should be based on facts, not feelings. You’d be more accurate with your recommendations if you follow up on the fecal samples and state that it’s safe to downgrade to BSL1 at 7 dpi. That should be, at a minimum, discussed!

Response:  The sentence was rephrased.  However, given that our authorship includes practicing clinicians as well as experienced virologists, we believe that it is appropriate to state what is “clinically relevant” based upon our experience, observations, and the data.

  1. You should also discuss that these recommendations only apply to mice infected with Ad5 – and that more studies are needed to make this applicable to other replication-competent Ads.

Response:  These points were added to the discussion section per the reviewer’s suggestion.

  1. Add a few words on the limitations of your study to the Discussion.

Response:  Consideration of the limitations were added to the discussion section per the reviewer’s suggestion.

  1. Please also provide an author contribution statement

Response:  This was added per the reviewer’s suggestion.

Reviewer #2:  This short manuscript examines the extent to which adenoviruses are shed after inoculation of lab mice, with a view to determining the level of biosafety required for these sorts of study.  The experiments appear to have been carefully carried out and give sensible results which should be of interest to the scientific community.  A couple of minor criticisms which should be addressed.

  1. How many mice were used for each time point in Figures 1 and 3?

Response:  A total of 15 mice were utilized in the experiment.  The results/methods were updated to reflect these numbers.

  1. In figure 1 for some time points it is difficult to distinguish between statistical * and the data symbol -this should be clarified.

Response:  The figure was updated to improve clarity and readability.

  1. The authors might speculate on reasons for the marked differences between the No tumor and Tumor results in some panels of Figure 1.

Response: 

All samples were collected, however, in some samples, the amount od DNA was not detectable / below the threshold, which was based on the combined baseline (day 0) values as we explained above and in the revised manuscript.

Reviewer 2 Report

This short manuscript examines the extent to which adenoviruses are shed after inoculation of lab mice, with a view to determining the level of biosafety required for these sorts of study.

The experiments appear to have been carefully carried out and give sensible results which should be of interest to the scientific community.

A couple of minor criticisms which should be addressed:

How many mice were used for each time point in Figures 1 and 3?

In figure 1 for some time points it is difficult to distinguish between statistical * and the data symbol -this should be clarified.

The authors might speculate on reasons for the marked differences between the No tumor and Tumor results in some panels of Figure 1.
